# How soon should patients be eligible for differentiated service delivery models for antiretroviral treatment? Evidence from a retrospective cohort study in Zambia

Lise Jamieson [1,2] Sydney Rosen [1,3] Bevis Phiri,[4] Anna Grimsrud [5] Muya Mwansa,[6] Hilda Shakwelele,[4] Prudence Haimbe,[4] Mpande Mukumbwa-Mwenechanya,[7] Priscilla Lumano-Mulenga,[6] Innocent Chiboma,[6] Brooke E Nichols [1,2,3]

Deceased 11 December 2022

For numbered affiliations see end of article.

**Correspondence to**
Dr Sydney Rosen;
sbrosen@bu.edu

## ABSTRACT

**Objectives** Patient attrition is high the first 6 months after antiretroviral therapy (ART) initiation. Patients with <6 months of ART are systematically excluded from most differentiated service delivery (DSD) models, which are intended to support retention. Despite DSD eligibility criteria requiring ≥6 months on ART, some patients enrol earlier. We compared loss to follow-up (LTFU) between patients enrolling in DSD models early with those enrolled according to guidelines, assessing whether the ART experience eligibility criterion is necessary.

**Design** Retrospective cohort study using routinely collected electronic medical record data.

**Setting**

**Participants** Adults (≥15 years) who initiated ART between 1 January 2019 and 31 December 2020.

**Outcomes** LTFU (>30 days late for scheduled visit) at 18 months for 'early enrollers' (DSD enrolment after <6 months on ART) and 'established enrollers' (DSD enrolment after ≥6 months on ART). We used a log-binomial model to compare LTFU risk, adjusting for age, sex, location, ART refill interval and DSD model.

**Results** For 6340 early enrollers and 25 857 established enrollers, there were no differences in sex (61% female), age (median 37 years) or location (65% urban). ART refill intervals were longer for established versus early enrollers (72% vs 55% were given 4–6 months refills). LTFU at 18 months was 3% (192 of 6340) for early enrollers and 5% (24 646 of 25 857) for established enrollers. Early enrollers were 41% less likely to be LTFU than established patients (adjusted risk ratio 0.59, 95% CI 0.50 to 0.68).

**Conclusions** Patients enrolled in DSD after <6 months of ART were more likely to be retained than patients established on ART prior to DSD enrolment. A limitation is that early enrollers may have been selected for DSD due to providers' and patients' expectations about future retention. Offering DSD models to ART patients soon after ART initiation may help address high attrition during the early treatment period.

**Trial registration number** NCT04158882.

## STRENGTHS AND LIMITATIONS OF THIS STUDY

⇒ Our analysis used data from Zambia's national electronic medical record system, with records from the entire national HIV treatment cohort over 4 years (2018–2021) in all 10 provinces.

⇒ We report observed outcomes for more than 6000 antiretroviral therapy (ART) clients who enrolled in differentiated service delivery (DSD) models after less than 6 months of experience on ART.

⇒ The results reflect large-scale, routine programme implementation, rather than clinical trial settings.

⇒ A key limitation is the assumption that patients who were enrolled in DSD models after less than 6 months on ART were selected based on an expectation of good future adherence.

⇒ A further limitation is the potential bias if facilities with better-than-average retention rates were more likely to allow early DSD model enrolment; the results may reflect differences in the quality of service as opposed to the relationship between duration on ART before DSD enrolment and retention in care.

## INTRODUCTION

A critical step towards achieving universal coverage of antiretroviral therapy (ART) for HIV is to support lifelong patient retention in ART programmes. Data from sub-Saharan Africa (SSA), where some 70% of the world's ART patients reside, continue to indicate insufficient retention on ART,[1] with about a fifth of all patients lost to care 5 years after treatment initiation.[2] A patient's first 6 months after initiation are a high risk period for attrition: a Zambian study showed the rates of loss to follow-up to be fourfold higher in the first 6 months of ART treatment compared with the period between 6 months and 3.5 years thereafter.[3]



Since 2016, the WHO has recommended differentiated service delivery (DSD) for HIV treatment.[4] DSD models such as facility-based individual 'fast-track' medication pick-up and community-based ART refills can increase access and remove barriers to care by adjusting the cadre of providers, location of service delivery, frequency of interactions with the healthcare system and/or types of services offered to support long-term retention of people established on HIV treatment.[5] A recent systematic review reporting on the outcomes of patients in DSD models in SSA found that retention in care of those in DSD models was generally within 5% of that for conventional care.[6] In Zambia, several DSD models have shown to have similar rates of retention as conventional care 12 months after DSD model entry.[7 8] The Varying Intervals of Antiretroviral Medication Dispensing to Improve Outcomes for HIV (INTERVAL) trial, a cluster-randomised, non-inferiority trial conducted in Malawi and Zambia, found that 6-month ART dispensing was non-inferior in terms of 12-month retention compared with standard of care.[8] DSD models have consistently been found to save substantial time and money for the patients themselves, and satisfaction with the models among both providers and patients has been high.[8–10]

A major limitation to the scale-up of DSD models to date has been the eligibility criteria which limit enrolment to patients who are 'stable' or 'established on treatment', which is defined as patients who (1) are on first-line ART regimens; (2) have been on ART for at least 6 or 12 months; and (3) have a recent, documented suppressed viral load.[8 11–13] Until April 2021, the WHO's definition of 'established' included at least 12 months of ART experience; new guidelines require at least 6 months on ART for DSD model eligibility.[14] Patients who are newly initiated on ART are thus systematically excluded from stable-patient-specific DSD models and from the benefits they offer. In the previously cited INTERVAL trial in Malawi and Zambia, 10% of all patients were excluded due to having initiated ART less than 6 months prior.[15] For patients not eligible for DSD models, guidelines typically require frequent visits to the healthcare facility and medication dispensing intervals of no more than 3 months.[16] In Zambia, all care is differentiated and dependent on the needs of the patient,[11] but currently there is no evidence on the outcomes of patients with <6 months of ART experience who enrol into DSD models that are typically reserved for stable patients.

Despite existing guidelines limiting DSD eligibility based on time on ART, in practice patients who do not meet guideline-recommended criteria are sometimes enrolled in DSD models for stable patients due to provider decision, error or patient request. To understand how such patients who are referred early to DSD models fare when participating in DSD models designed for those established on treatment, we analysed routinely collected medical record data from Zambia to compare the rates of retention among patients enrolled into DSD models earlier than guidelines recommend with retention among those who met all eligibility criteria.

## METHODS
### Study population and outcomes
We conducted a retrospective cohort study with data extracted in October 2021 from SmartCare, Zambia's national electronic medical record system.[17] We extracted data for patients aged 15 years or older reported to have initiated ART between January 2019 and December 2020 at any of 692 health facilities across all 10 provinces. Zambian policy guidelines for this period required patients to be stable on ART before they are considered for DSD enrolment, with stability defined in the 2018 consolidated ART guidelines[11 12] as on ART for at least 6 months.

We defined patients who enrolled into a DSD model with <6 months of ART as 'early enrollers', with a comparison group of patients who enrolled into a DSD model with ≥6 months of ART defined as 'established enrollers'. Patients on second-line ART (defined as those dispensed protease inhibitors such as lopinavir, atazanavir or ritonavir) were excluded from this analysis as they are already known to be at high risk of attrition.[18 19] For both early and established enrollers, we assessed loss to follow-up at 18 months post-ART initiation, with loss to follow-up defined as patients who were reported as 'lost to follow-up' or 'inactive' in the SmartCare database between 15 and 21 months after ART initiation date. 'Inactive' was defined as having missed a scheduled visit by more than 30 days. Rates of loss to follow-up were calculated for early and established enrollers and stratified by DSD model type and ART dispensing duration. DSD models, which had multiple names in the SmartCare database, were grouped into the following categories: (1) adherence groups (community adherence groups, rural/urban adherence groups); (2) extended clinic hours (DSD models designed for clinic access before/after hours or weekends, including scholar models); (3) fast-track (procedures to accelerate dispensing at clinics); (4) home ART delivery; (5) multimonth dispensing (MMD); and (6) community pick-up point (central dispensing units, community retail pharmacies, community ART distribution points, health posts, mobile ART distribution models) (table 1). These six DSD models were defined in our analysis to be mutually exclusive; patients could only be enrolled in a single model.

### Statistical analysis
We described the demographics of our study population using descriptive statistics. We compared risk of loss to follow-up between early enrollers and established enrollers, and Wilson's score interval was used to calculate 95% CIs around proportions. We used a log-binomial regression to calculate risk ratios for loss to follow-up, adjusting for age, sex, urban/rural status, DSD model

**Table 1** Differentiated service delivery models for HIV treatment in use in Zambia during the study period

| Category | Models in the category | Description |
|---|---|---|
| Adherence groups | Community adherence groups. | Patient groups, consisting of ±6 members, meeting at an agreed time every 1–3 months. The groups are managed by the patients themselves and usually meet outside of the health facility. Members collect ART at clinical appointments for other members in a rotating fashion.[7] |
| | Rural and urban adherence groups/clubs. | Patient groups, consisting of 20–30 members, meeting at an agreed time every 2–3 months. Groups are often facilitated by the same healthcare worker or facility-based volunteer, also providing prepackaged ART.[7] |
| Community pick-up point | Central dispensing units. | A centralised model for ART distribution where medication is packed at a centrally located hub and distributed to patients at multiple approved pick-up points. Clinic visits occur every 6 months at the health facility.[11] |
| | Community ART distribution points, community retail pharmacies and health posts. | ART refills are provided to patients outside of health facilities, for example, schools, churches, community centres, community retail pharmacies and health posts.[11] |
| | Mobile ART distribution models. | A clinical outreach team linked to a facility does 3-monthly clinical assessments at community distribution points. This model is usually used for hard-to-reach areas.[11] |
| Extended clinic hours | Before/after-hours models, weekend models and scholar models. | These models allow patients to have a clinic visit and collect their ART outside the conventional operation times at the facility (early mornings, evenings and over weekends). These are beneficial to patients with competing priorities (eg, school or employment). |
| Fast-track | Fast-track. | A model that typically involves a separate, shorter queue to dispense ART to stable patients, allowing for a quick patient visit when a clinic visit is not required.[23] |
| Home ART delivery | Home ART delivery. | Trained community health workers linked to facilities conduct home visits to deliver ART, conduct health screening, monitor adherence and refer patients as required.[7] |
| Multimonth dispensing | Multimonth dispensing. | Facility-based model in which the primary goal is to dispense medications for more than 1 month (usually 6 months). Dispensing is typically done during a clinic facility-based visit. |

ART, antiretroviral therapy.

type and ART dispensing duration. Analyses were also stratified by DSD model type and ART dispensing duration. Further, we also conducted an age-stratified analysis and a subanalysis restricted to facilities with a higher proportion of early enrollers, with results shown in the online supplemental material.

### Patient and public involvement
Patients and the public were not involved in the design and conduct of this research.

## RESULTS
### Study populations
The full SmartCare data set included 1 520 125 unique patients on ART over 2018–2021, of whom 32 197 had enrolled into a DSD model after ART initiation and had an 18-month outcome reported within the window of 15–21 months (figure 1). Of these, 6340 patients were reported to have been enrolled in DSD models <6 months after ART initiation during the study period (early enrollers). The remaining 25 857 patients comprised the comparison

group of established enrollers. For early enrollers, the median time enrolled in a DSD model at the time of outcome evaluation was 14.7 months (IQR 13.0–16.5); majority (81%, n=20 856) of established enrollers were on DSD models at outcome evaluation at a median of 5.8

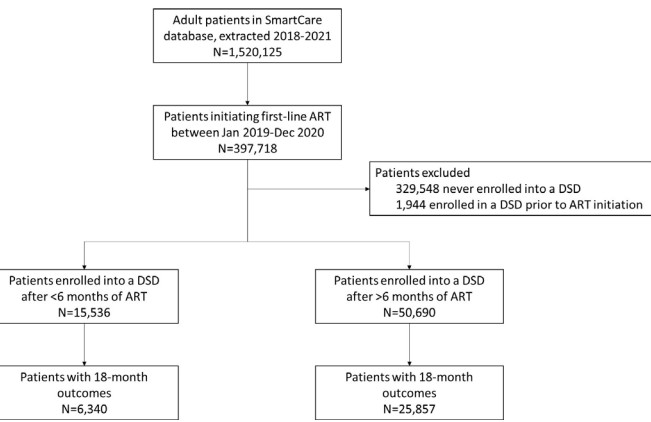

**Figure 1** Flow diagram depicting study population. ART, antiretroviral therapy; DSD, differentiated service delivery.

**Table 2** Demographics of patients enrolled in DSD models

| Variable | | Early enrollers of DSD models (n=6340) | Established enrollers of DSD models (n=25857) |
|---|---|---|---|
| Age in years, median (IQR) | | 36 (29–44) | 37 (29–44) |
| Age group, n (%) | 15–24 | 727 (11) | 2589 (10) |
| | 25–34 | 2069 (33) | 8346 (32) |
| | 35–49 | 2658 (42) | 11424 (44) |
| | 50+ | 885 (14) | 3487 (13) |
| Sex, n (%) | Female | 3914 (62) | 15666 (61) |
| | Male | 2426 (38) | 10191 (39) |
| Location, n (%) | Rural | 2501 (39) | 9078 (35) |
| | Urban | 3839 (61) | 16779 (65) |
| Year of ART initiation, n (%) | 2019 | 2897 (46) | 17346 (67) |
| | 2020 | 3443 (54) | 8511 (33) |
| DSD type, n (%) | Adherence groups | 149 (2) | 508 (2) |
| | Community pick-up points | 671 (11) | 1461 (6) |
| | Extended clinic hours | 85 (1) | 97 (<1) |
| | Fast-track | 979 (15) | 6266 (24) |
| | Home ART delivery | 355 (6) | 973 (4) |
| | Multimonth dispensing | 4101 (65) | 16552 (64) |
| ART months dispensed, n (%) | <2 | 636 (10) | 1476 (6) |
| | 3 | 2197 (35) | 5688 (22) |
| | 4–6 | 3507 (55) | 18679 (72) |
| Outcome year, n (%) | 2020 | 2863 (45) | 17283 (67) |
| | 2021 | 3477 (55) | 8574 (33) |
| Months on ART at outcome, median (IQR) | | 17.9 (16.4–19.5) | 18.4 (16.7–19.8) |
| On DSD at outcome, n (%) | Yes | 6340 (100) | 20856 (81) |
| | No | 0 (0) | 5001 (19) |
| Months on DSD at outcome, median (IQR) | | 14.7 (13.0–16.5) | 5.8 (2.9–8.9) |
| Patient outcomes by 18 months after ART initiation, n (%) | On treatment | 6133 (97) | 24646 (95) |
| | Died | 11 (<1) | 31 (<1) |
| | Lost to follow-up | 192 (3) | 1169 (5) |
| | Stopped ART | 4 (<1) | 10 (<1) |
| | Stopped DSD | 0 (0) | 1 (<1) |

ART, antiretroviral therapy; DSD, differentiated service delivery.

months (IQR 2.9–8.9) (table 2). Early enrollers and established enrollers were similar with respect to age, sex and urban/rural location. Across both groups, the median age was 37 years (IQR 29–44), majority (61%, 19580 of 32197) were female and most patients resided in urban settings (64%, n=20618).

Most patients were enrolled in either the MMD DSD model (65% (n=4101) of early enrollers and 64% (n=16552) of established enrollers) or the fast-track model (15% (n=979) of early enrollers and 24% (n=6266) of established enrollers) (table 1). Among early enrollers, around half (55%, n=3477) were dispensed 4–6 months of ART at their most recent ART pick-up, 35% (n=2197) were dispensed 3 months of ART and 10% (n=636) were

dispensed <2 months of ART. Established enrollers had slightly longer dispensing intervals, with 72% (n=18679) dispensed 4–6 months of ART, 22% (n=5688) dispensed 3 months of ART and 6% (n=1476) dispensed <2 months of ART (table 1).

### Outcomes
Early enrollers had a slightly lower rate of loss to follow-up (3.0%, 95% CI 2.6% to 3.5%) compared with established enrollers (4.5%, 95% CI 4.3% to 4.8%) (table 3). Early enrollers experienced similar or lower loss to follow-up rates than established enrollers across nearly all differentiated models of care. The exception was extended clinic hours: early enrollers enrolled in the extended clinic

**Table 3** Relative risk of loss to follow-up at 18 months post-ART initiation for early enrollers of DSD models

| | Proportion of patients lost to follow-up at 18 months, % (95% CI) (n/N) | | Unadjusted risk ratio (95% CI) | Adjusted risk ratio* (95% CI) |
|---|---|---|---|---|
| | **Early enrollers** | **Established enrollers** | | |
| All patients | 3.0 (2.6 to 3.5) (192/6340) | 4.5 (4.3 to 4.8) (1169/25 857) | 0.67 (0.57 to 0.78) | 0.59 (0.50 to 0.68) |
| Stratification: DSD model | | | | |
| Adherence groups | 2.7 (1.0 to 6.7) (4/149) | 3.1 (1.9 to 5.1) (16/508) | 0.85 (0.25 to 2.29) | 0.79 (0.23 to 2.12) |
| Community pick-up points | 4.5 (3.1 to 6.3) (30/671) | 3.3 (2.5 to 4.3) (48/1461) | 1.36 (0.86 to 2.12) | 1.30 (0.81 to 2.03) |
| Extended clinic hours | 10.6 (5.7 to 18.9) (9/85) | 8.2 (4.2 to 15.4) (8/97) | 1.28 (0.51 to 3.27) | 1.19 (0.43 to 3.34) |
| Fast-track | 3.4 (2.4 to 4.7) (33/979) | 3.6 (3.2 to 4.1) (227/6266) | 0.93 (0.64 to 1.31) | 0.74 (0.50 to 1.05) |
| Home ART delivery | 1.4 (0.6 to 3.3) (5/355) | 6.3 (4.9 to 8) (61/973) | 0.22 (0.08 to 0.50) | 0.18 (0.06 to 0.41) |
| Multimonth dispensing | 2.7 (2.3 to 3.2) (111/4101) | 4.9 (4.6 to 5.2) (809/16 552) | 0.55 (0.45 to 0.67) | 0.51 (0.41 to 0.61) |
| Stratification: ART dispensing duration | | | | |
| <2 months | 4.1 (2.8 to 5.9) (26/636) | 10.6 (9.1 to 12.2) (156/1476) | 0.39 (0.25 to 0.57) | 0.40 (0.26 to 0.59) |
| 3 months | 3.5 (2.8 to 4.4) (77/2197) | 5.3 (4.8 to 5.9) (303/5688) | 0.66 (0.51 to 0.84) | 0.64 (0.49 to 0.81) |
| 4–6 months | 2.5 (2.1 to 3.1) (89/3507) | 3.8 (3.5 to 4.1) (709/18 679) | 0.67 (0.54 to 0.83) | 0.67 (0.53 to 0.82) |

*Model adjusted for age, sex, location, ART dispensing duration and DSD model type.
ART, antiretroviral therapy; DSD, differentiated service delivery.

hours model had a similar rate of loss to follow-up as established enrollers (10.6% (95% CI 5.7% to 18.9%) vs 8.2% (95% CI 4.2% to 15.4%), respectively). Across both early and established enrollers, longer dispensing periods were associated with lower rates of loss to follow-up, which increased from 2.5%–3.8% for 4–6 months dispensing to 3.5%–5.3% for 3 months dispensing, to 4.1%–10.6% for <2 months dispensing (table 3). Early enrollers with <2 months dispensing had a lower rate of loss to follow-up than did established enrollers (4.1% (95% CI 2.8% to 5.9%) vs 10.6% (95% CI 9.1% to 12.2%)).

In an analysis adjusting for age, sex, location, ART dispensing duration and DSD model type, early enrollers in all DSD model types and dispensing durations were 41% less likely to be lost to follow-up than established enrollers (adjusted risk ratio (aRR) 0.59, 95% CI 0.50 to 0.68) (table 3). The reduced adjusted risk of being lost to follow-up was similar for patients in adherence groups (aRR 0.79, 95% CI 0.23 to 2.12), MMD (aRR 0.51, 95% CI 0.41 to 0.61), home ART delivery (aRR 0.18, 95% CI 0.06 to 0.41) and fast-track (aRR 0.74, 95% CI 0.50 to 1.05) models. Early enrollers had a statistically insignificant increased risk of being lost to follow-up in the community pick-up point (aRR 1.30, 95% CI 0.81 to 2.03) and

extended clinic hours (aRR 1.19, 95% CI 0.43 to 3.34) models compared with established enrollers.

An age-stratified analysis produced similar results to the main analysis, with early enrollers in each age group being less likely to be lost to follow-up than established enrollers in the same age group. However, the effect of earlier enrolment in DSD on reduced loss to follow-up appeared less pronounced in patients on 4–6 months ART dispensing for those aged 25–49 years (online supplemental figure S1). In facilities where a larger proportion of all DSD patients enrolled in DSD models early, the trend towards early enrollers performing better persisted with respect to loss to follow-up compared with the outcomes for established enrollers (online supplemental figure S2).

## DISCUSSION
In nearly all of SSA, DSD model eligibility criteria require that patients be on ART for a minimum of 6 months (and in some countries a minimum of 12 months) prior to DSD model enrolment.[20] We present a novel analysis from Zambia highlighting good outcomes when newly initiated ART patients (those with less than 6 months of

ART experience) are referred early to DSD models. Those referred early to DSD appear to have good outcomes across different DSD models and age categories.

Our data begin to fill in a gap in the evidence base on the validity of time on treatment as an eligibility criterion for DSD models. Because few if any countries permit DSD model enrolment for new initiators, little evidence on their experience in DSD models has been available until now. To date, most reports on DSD outcomes have been limited to people who have spent a significant amount of time on ART prior to DSD model enrolment. In the previously mentioned INTERVAL trial, for example, participants had been on ART for a median of roughly 5 years at DSD model entry, while patients in a trial of MMD in adherence clubs in South Africa had a median duration on ART of 7.3 years at baseline.[21]

While ART patients in Zambia have historically been lost to follow-up at high rates in the first few months after ART initiation,[3] in our DSD patient population this was less likely to be the case. Our results provide evidence to support the recent revision of the WHO guidelines that reduce time on ART from 12 months to 6 months on treatment as part of the definition of 'established' on ART.[14] These findings offer reassurance and evidence to countries that have expanded eligibility as they scale up DSD models,[20 22] particularly to support uninterrupted access to HIV treatment during the COVID-19 pandemic, that earlier referral to DSD is possible without compromising patient care. Even if many, or most, of the patients in our 'early enrolment' sample were selected deliberately because they were considered at low risk of loss to follow-up, our results demonstrate that early eligibility for DSD models should be considered for at least some patients before they reach 6 months on ART.

Loss to follow-up at 18 months after ART initiation for early and established enrollers averaged 1%–11% for all six categories of DSD models studied. We did not observe any programmatically important differences by model or ART experience prior to model enrolment. Where a programmatically important difference did arise, in contrast, was in dispensing intervals. Regardless of how long a patient had been on ART at DSD model enrolment, patients who received ≤2 months of medications at a time were more likely to be lost to follow-up than patients who received either 3 months or 4–6 months of medications. This likely reflects providers' assessments of patients' ability to remain on treatment and/or clinical condition. Those regarded as being at higher risk of attrition are asked to come to the clinic for medication refills more often so that they can be monitored and supported more closely. Ironically, difficulty in accessing the clinic may be the very reason that some patients are at high risk of attrition. For these patients, insisting on shorter refill durations may simply exacerbate whatever challenges they face.

There were several limitations to our analysis. First, we cannot explain why some patients were enrolled in DSD models before reaching 6 months on ART. As noted above, we assume that patients with <6 months on ART in our sample were not offered DSD model enrolment at random. If providers made accurate clinical decisions about individual patients' risks of attrition, patients in our 'early enrolment' cohorts could over-represent patients thought to have low attrition risk. To achieve the results we found, providers would have had to make these decisions correctly at multiple sites across the entire country. If this is the case, our data suggest that the healthcare workers responsible for enrolling patients into DSD models can successfully identify those who will do well with early enrolment. At the same time, if the early enrollers in our data set do comprise patients at lower risk of loss to follow-up, then our results likely underestimate the true rate of loss to follow-up that would occur if early DSD enrolment were to be broadly available, without the benefit of provider selection.

A second limitation is that our data set included only patients reported in the electronic medical record system to have enrolled in a DSD model. It is possible that some patients not in DSD models may be recorded as enrolled and some who were enrolled may have been missed. Third, bias could occur if facilities with better-than-average retention in care were also more likely to allow early DSD model enrolment. In this case, our results may reflect differences in facility quality as well as enrolment timing. An analysis restricted to facilities with >20% early DSD enrolment showed an even lower risk of loss to follow-up among patients enrolled early into DSD models, however, compared with patients with >6 months of ART at DSD entry.

Despite these limitations, our analysis demonstrates that patients on ART for less than 6 months who are enrolled in existing DSD models can be successfully retained in care and may even fare better than those left in conventional care and only initiate DSD models greater than 6 months after ART initiation. It is likely that not all patients are ready for less intensive DSD models in their first half-year or year on treatment, but some clearly are. Since DSD models have been shown to be beneficial to patients and in some cases to providers, offering enrolment to newly initiating ART patients may improve ART programmes in general. Future research should look more closely at which patients can be enrolled early and which models of care serve these patients best.

## CONCLUSION

The current policy for DSD model eligibility criteria in Zambia, as in other countries, requires a minimum of 12 months of ART before a patient is considered for DSD enrolment, and more recently a minimum of 6 months of ART. In order to change the guidelines to allow DSD enrolment sooner after ART initiation (ie, 6 months or less), large-scale observational evidence, implementation research or trial data demonstrating good patient outcomes among those who enrol in DSD models <6 months post-ART initiation would be required. This

analysis therefore provides a critical first step towards the reassessment of the delayed DSD enrolment policies and signals that further research needs to be conducted in other SSA countries to evaluate patient outcomes for early DSD model enrolment.

**Author affiliations**
¹Health Economics and Epidemiology Research Office, University of the Witwatersrand Faculty of Health Sciences, Johannesburg, South Africa
²Department of Medical Microbiology, Amsterdam University Medical Centre, Amsterdam, the Netherlands
³Department of Global Health, Boston University School of Public Health, Boston, Massachusetts, USA
⁴Clinton Health Access Initiative, Lusaka, Zambia
⁵International AIDS Society, Cape Town, South Africa
⁶Ministry of Health, Lusaka, Zambia
⁷Implementation Science Unit, Center for Infectious Disease Research in Zambia, Lusaka, Zambia

**Contributors** LJ, BN, SR and AG conceptualised the study. BP, HS, PH, MM-M, PL-M and IC curated data for the study. BP, HS, PH and MM-M provided supervision of the study. LJ led the data analysis and drafted the paper along with BN, SR and AG. SR is the guarantor for this study. All authors contributed to data interpretation and critically reviewed a revised draft of the manuscript. All authors have read and approved the final manuscript.

**Funding** Funding for the study was provided by the Bill & Melinda Gates Foundation through OPP1192640 to Boston University.

**Competing interests** None declared.

**Patient and public involvement** Patients and/or the public were not involved in the design, or conduct, or reporting, or dissemination plans of this research.

**Patient consent for publication** Not required.

**Ethics approval** This study involves human participants and was approved by ERES Converge IRB (Zambia) (protocol number 2019-Sep-030), the Human Research Ethics Committee (Medical) of the University of Witwatersrand (protocol number M190453) and Boston University IRB (H-38823) for use of data with a waiver of consent. All data were routinely collected, retrospective medical record data.

**Provenance and peer review** Not commissioned; externally peer reviewed.

**Data availability statement** Data may be obtained from a third party and are not publicly available. The data are owned by the Zambian Ministry of Health and their use was approved by the Human Research Ethics Committee (University of Witwatersrand, Johannesburg, South Africa) and ERES Converge IRB (Zambia). All relevant data are included in the paper and supplementary material. Full data are available upon approval from the Zambian Ministry of Health and appropriate ethics committees.

**ORCID iDs**
Lise Jamieson http://orcid.org/0000-0003-2354-4580
Sydney Rosen http://orcid.org/0000-0002-6560-2964
Anna Grimsrud http://orcid.org/0000-0002-1199-8377
Brooke E Nichols http://orcid.org/0000-0003-4682-4999

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
