## [Reviewer comments · BMJ Open]

ARTICLE DETAILS

TITLE (PROVISIONAL)	How soon should patients be eligible for differentiated service delivery models for antiretroviral treatment? Evidence from a retrospective cohort study in Zambia
AUTHORS	Jamieson, Lise; Rosen, Sydney; Phiri, Bevis; Grimsrud, Anna; Mwansa, Muya; Shakwelele, Hilda; Haimbe, Prudence; Mukumbwa-Mwenechanya, Mpande; Lumano-Mulenga, Priscilla; Chiboma, Innocent; Nichols, Brooke

VERSION 1 – REVIEW

REVIEWER	Shenoi, Sheela Yale University School of Medicine, Section of Infectious Diseases
REVIEW RETURNED	13-Jun-2022

GENERAL COMMENTS	This report is an excellent opportunity to evaluate previously collected routine data. • What is the interval between ART pickups in the Community ART distribution points, community retail pharmacies, health posts, and the central dispensing units?• 6340 (25%) of initiators were placed in the DSD early. This seems high for a setting that runs on protocols for ART initiation. It is somewhat surprising that HCWs ignore the protocol in such a high proportion of initiators. What reasons could explain why such a high proportion of patients were initiated on ART outside the standard protocol? While the authors appropriately recognize this limitation that potentially undermines the results, the subanalysis is reassuring.• Please include p-values for Table 1 – need to see the significant differences between early and established PWH• How were the criteria identified for the subanalysis? ≥20% of patients had early enrollers, and ii) at least 100 patients across both groups (early enrollers and established enrollers). Despite the major limitation that patients may have been placed in the DSD protocol too early and therefore may undermine the results, the study does offer a subanalysis showing similar results, and does take advantage of a unique opportunity to evaluate the role DSD from the time of initiation and provides useful information to the literature.
---

REVIEWER	Mpofu, Mulamuli Chemonics International
REVIEW RETURNED	30-Jun-2022

GENERAL COMMENTS	Overall Comments The authors focused on a critical component of the HIV response. With several countries having made lot of progress on putting patients on ART, the focusing is now focusing on ensuring good outcomes which include retaining patient in care. The manuscript is well written. Major revisions: Methods Line 133- 144: Authors should have assessed and compared LFTU 18 months since enrollment into DSD models and not enrollment into ART. As the results show, using the current approach, the established enrollers would have been on DSD for a short period (median is 4.5 months) and technically still adapting and establishing themselves under DSD while the early enrollers (median 14 months) are well established on DSD by the time of the study. Since the exposure of interest is being on DSD, authors should compare LFTU (the outcomes), 18 months after enrollment into DSD . Discussion Table 3: Looking at the different DSD models, it looks like only home ART delivery and MMD (though not clear if its MMD3,4,5 or 6) were significantly associated with low LFTU. Based on this, the authors should then exercise caution when arguing for DSD as some of the DSD models had worse LFTU rates amongst early enrollers than established enrollers. Minor Revisions Title The authors should consider modifying the title to align with the results. As is, the manuscript is not showing any evidence of when patients should be initiated on DSD and there are mixed results depending on the DSD model. I suggest the title to be modified to: How soon should patients be eligible for differentiated service delivery models for antiretroviral treatment? Analysis of program data from Zambia Abstract Line 29 – replace 6 with six Line 31 – replace “which are intended to reduce attrition” with “which can reduce attrition” because DSD models are not necessarily for reducing attrition only but also to decongest health facilities, offer convenience to patients etc. Line 33--- add “national” before guidelines. Line 34 --- add “for DSD enrollment” after necessary. Line 52 – Suggest remove the limitation because it is based on an assumption. Line 53 ---The sentence starting with “Offering DSD” should be removed because it is vague. “At least some” is not measurable since the minimum number is unknown. Introduction Line 99—add ‘to scale-up’ after limitation. Line 99 – delete “model” after limit. Please revise the entire sentence from line 99 – 101 because it is confusing. Line 103—I suggest removal of sentence which starts with “Patients” because it gives an impression that those on treatment
---

	for less than 6 months are deliberately excluded when they could be stable. The definition of stable patient is very clear, continues to be revised based on scientific and professional judgement and is meant to standardize practice. Line 114 – I suggest removal of the first part of the sentence starting with “to begin” and use text starting after the comma in line 116. Please provide data on proportion of patients on DSD in Zambia. Methods Results Line 163: How many of the 1,520,125 unique patients had initiated ART between 01/01/2019 - 31/12/2020? Table 2: Clarify if there were no patients enrolled in multiple DSD models e.g. Fast track and MMD? Outcomes While the interest of authors is to argue against eligibility for DSD enrollment, it would have been good to also show LTFU rates amongst those who were never enrolled on DSD. That helps to sustain the arguments for DSD and then focus on when it should be applied. Discussion Line 220: Remove the word “novel” and “good”. Line 224- remove “begin to” Line 2343– indicate what this high rates are. Limitations: The failure to compare urban areas and rural areas is another limitation which the authors should acknowledge. Are the models effective the same way in both areas? Title: The authors should modify the title to align with the results. As is, the manuscript is not showing any evidence of when patients I suggest the title to be modified to: How soon should patients be eligible for differentiated service delivery models for antiretroviral treatment? Analysis of program data from Zambia Introduction
--	--

VERSION 1 – AUTHOR RESPONSE

Reviewer 1

Dr. Sheela Sheno, Yale University School of Medicine

Comments to the Author:

This report is an excellent opportunity to evaluate previously collected routine data.

Thank you.

- What is the interval between ART pickups in the Community ART distribution points, community retail pharmacies, health posts, and the central dispensing units?

Thank you for this question. The ART refill duration varies between 1 and 6 months in all of these options, with 4–6 month multimonth dispensing being the most common (Table 2). In our analysis, we adjusted for the duration of ART refills when determining the risk ratios of loss to follow-up among early versus established enrollers (Table 3). We did not include dispensing intervals within each model in this paper because our focus here is on outcomes by time on ART at DSD model enrolment. Another manuscript that reports this information is currently under review.

- 6340 (25%) of initiators were placed in the DSD early. This seems high for a setting that runs on protocols for ART initiation. It is somewhat surprising that HCWs ignore the protocol in such a high proportion of initiators. What reasons could explain why such a high proportion of patients were initiated on ART outside the standard protocol? While the authors appropriately recognize this limitation that potentially undermines the results, the subanalysis is reassuring.

We agree, a surprisingly large proportion of our sample were “early enrollers” with less than six months on ART before referral to a DSD model. Two thirds of these (65%, Table 2) were in the multi-month dispensing model with six-monthly clinical visits. We do not have any data from this research to offer reasons as to why a quarter of patients were enrolled before they reached the required duration on ART. Our speculation is that clinicians are making judgements as to those who would benefit and could be trusted to receive early referral and longer ART dispensing and are selecting these patients to have early access. It is also possible that patients are aware of the option of six-month dispensing (and other DSD models) and are asking to enroll early. We have noted in the limitations section that we cannot explain the reasons for the observed patterns.

- Please include p-values for Table 1 – need to see the significant differences between early and established PWH

The results present baseline characteristics of patients. With respect to the reviewer’s concern about adding in p-values, we understand the concern, but we feel that baseline characteristics are the place to look for differences between the groups included and excluded in order to assess potential confounding. As confounding is a bias issue rather than a statistical issue, we are not trying to draw inferences to a larger population for which one would use statistical tests and p-values. Rather we want the reader to assess the potential impact of baseline differences on the outcomes analysed, a problem which deals with the actual differences observed, not whether or not they are statistically significant. The STROBE statement on reporting of observational data specifically notes that statistical significance should not be used for descriptive tables (ref: <http://www.plosmedicine.org/article/info%3Adoi%2F10.1371%2Fjournal.pmed.0040297>). Therefore, unless specifically requested to by the editor, we would prefer not to include significance testing and p-values.

- How were the criteria identified for the subanalysis? ≥20% of patients had early enrollers, and ii) at least 100 patients across both groups (early enrollers and established enrollers).

We assume that this comment refers to Figure S2 in the supplementary appendix. The criteria for creating the subanalysis data set were selected based on author judgment as to what might constitute meaningful thresholds for this subanalysis.

Despite the major limitation that patients may have been placed in the DSD protocol too early and therefore may undermine the results, the study does offer a subanalysis showing similar results, and does take advantage of a unique opportunity to evaluate the role DSD from the time of initiation and provides useful information to the literature.

We note that in this manuscript, patients being placed in the DSD protocol too early is an exposure, rather than a limitation. We appreciate that the Reviewer finds the information in the manuscript useful, however.

Reviewer 2

Dr. Mulamuli Mpofu, Chemonics International

Comments to the Author:

Overall Comments

The authors focused on a critical component of the HIV response. With several countries having made lot of progress on putting patients on ART, the focusing is now focusing on ensuring good outcomes which include retaining patient in care. The manuscript is well written.

Thank you very much.

Major revisions:

Methods

Line 133- 144: Authors should have assessed and compared LFTU 18 months since enrollment into DSD models and not enrollment into ART. As the results show, using the current approach, the established enrollers would have been on DSD for a short period (median is 4.5 months) and technically still adapting and establishing themselves under DSD while the early enrollers (median 14 months) are well established on DSD by the time of the study. Since the exposure of interest is being on DSD, authors should compare LFTU (the outcomes), 18 months after enrollment into DSD .

Thank you for this comment. We did consider this approach. The goal of HIV treatment programmes, however, is to support sustained retention in treatment and sustained viral suppression. Therefore, our outcome estimates these achievement of these goals (sustained retention in the HIV programme and viral suppression), regardless of the service delivery model (standard of care or DSD model). If we had compared outcomes 18 months after enrolment into DSD, rather than 18 months after ART initiation, as the reviewer suggests, our results would have captured the impact of DSD models in general, but not of early enrolment into DSD. The exposure of interest for this analysis is not simply being enrolled in DSD models, but rather early enrolment into DSD, prior to six months' experience on ART. We were interested in the impact of early enrolment into any DSD model on retention (or loss-to-follow up) and not the specific impact of specific DSD models on retention after patients are enrolled.

Discussion

Table 3: Looking at the different DSD models, it looks like only home ART delivery and MMD (though not clear if its MMD3,4,5 or 6) were significantly associated with low LFTU. Based on this, the authors should then exercise caution when arguing for DSD as some of the DSD models had worse LFTU rates amongst early enrollers than established enrollers.

Thank you for this feedback. The reviewer is correct that “home ART delivery” and “multi-month dispensing” both had statistically significant reduced proportions of loss to follow-up at 18 months in the early enrollers compared to the established enrollers. For the other models, however, there was no statistically difference in the risk ratio (adjusted or unadjusted) between early and established enrollers. This lack of a difference argues for allowing early enrollers into DSD models, as these models have other benefits for patients and providers beyond those reflected in retention and viral suppression (e.g. lower costs). We have taken care with our language throughout the manuscript not to overstate our findings or their generalizability, but have simply emphasized that for the patients observed, early enrolment into DSD models was at least as good as conventional care.

Minor Revisions

Title

The authors should consider modifying the title to align with the results. As is, the manuscript is not showing any evidence of when patients should be initiated on DSD and there are mixed results depending on the DSD model. I suggest the title to be modified to: How soon should patients be eligible for differentiated service delivery models for antiretroviral treatment? Analysis of program data from Zambia

Thank you – we have revised the title to include the study design as requested by the editors. The title is now, “How soon should patients be eligible for differentiated service delivery models for antiretroviral treatment? Evidence from a retrospective cohort study in Zambia”

Abstract

We thank the reviewer for these edits. Unfortunately, due to restrictions to word count, not all the suggested edits could be made.

Line 29 – replace 6 with six

Thank you – done.

Line 31 – replace “which are intended to reduce attrition” with “which can reduce attrition” because DSD models are not necessarily for reducing attrition only but also to decongest health facilities, offer convenience to patients etc.

Thank you – we have amended to this to read “to support retention”. It is true that DSD models have multiple goals, with supporting retention being one of them.

Line 33--- add “national” before guidelines.

With tight limits on the abstract word count, this edit was not made as it is implied.

Line 34 --- add “for DSD enrollment” after necessary.

With limits to the word count, this edit was not made as this is implied in the existing sentence.

Line 52 – Suggest remove the limitation because it is based on an assumption.

We would like to keep this limitation in the abstract. While it is indeed an assumption, it is a well justified one and likely constitutes the most important constraint in interpreting the analysis. We have always in the past been asked to include the most important limitation(s) in the abstract and will retain this here unless the editors prefer that we remove or revise it.

Line 53 ---The sentence starting with “Offering DSD” should be removed because it is vague. “At least some” is not measurable since the minimum number is unknown.

Thanks for this feedback. We chose the word “offering” as DSD is intended to be client-centered and enrolment in a DSD model is supposed to be something that the facility offers its clients, not requires of them. We have deleted “at least some.”

Introduction

Line 99—add ‘to scale-up’ after limitation. Line 99 – delete “model” after limit. Please revise the entire sentence from line 99 – 101 because it is confusing.

Thank you – we have revised this sentence now. It now reads, “A major limitation to the scale-up of DSD models to date has been eligibility criteria that limit enrollment to patients who are “stable” or “established on treatment, which is defined as patients who: i) are on first-line ART regimens; ii) have been on ART for at least 6 or 12 months; and iii) have a recent, documented suppressed viral load.^{8,11–13”}

Line 103—I suggest removal of sentence which starts with “Patients” because it gives an impression that those on treatment for less than 6 months are deliberately excluded when they could be stable. The definition of stable patient is very clear, continues to be revised based on scientific and professional judgement and is meant to standardize practice.

In fact this impression is correct—those on treatment for less than 6 months are deliberately excluded, as they do not meet the WHO definition of an “established” patient. We have therefore kept the sentence as is.

Line 114 – I suggest removal of the first part of the sentence starting with “to begin” and use text starting after the comma in line 116.

We have revised the sentence and removed “begin to”.

Please provide data on proportion of patients on DSD in Zambia.

The proportion of patients on DSD scale-up in Zambia is reported in the first line of the results – with 32,197 patients in our cohort enrolled in a DSD model.

Methods

Results

Line 163: How many of the 1,520,125 unique patients had initiated ART between 01/01/2019 - 31/12/2020?

As shown in Figure 1, 397,718 patients initiated ART between January 2019 and December 2020. A large proportion of these were excluded from the analysis, however, because they never enrolled in a DSD model during our study period. We therefore did not think it necessary to specify this number in the text as well as in the figure. We can add it to the text if the editor prefers.

Table 2: Clarify if there were no patients enrolled in multiple DSD models e.g. Fast track and MMD?

This point is well taken. A sentence has been added to the methods to clarify. “These six DSD models were defined for our analysis to be mutually exclusive – patients could only be enrolled in a single model.”

Outcomes

While the interest of authors is to argue against eligibility for DSD enrollment, it would have been good to also show LTFU rates amongst those who were never enrolled on DSD. That helps to sustain the arguments for DSD and then focus on when it should be applied.

To clarify, we do not have an interest to argue for or against specific eligibility criteria but rather to report what we observed in our research. The objective of this analysis was to ascertain if there was a difference between outcomes for those who enrolled into DSD models with less than the guideline-recommended duration of experience on ART and outcomes for those who enrolled with at least the guideline-recommendation duration. We did this to provide evidence as to whether early enrolment (with <6 months’ experience on ART) could be successful and should be incorporated into guidelines or should instead be kept as a blanket exclusion criterion. We know there are important differences between those enrolled and those not enrolled into DSD models, with those enrolled more likely to have better treatment outcomes. Therefore, understanding the differences between those enrolled and not was beyond the scope of this analysis.

Discussion

Line 220: Remove the word “novel” and “good”.

We are not certain why these should be removed. This analysis was novel, as it contributed new evidence to understand the outcomes of those enrolled in DSD early after ART initiation. To our knowledge, an analysis like this has not been done before. Similarly, the outcomes in terms of retention after 18 months on ART were good—either as good as those achieved in conventional care or better, depending on the DSD model. If there is a specific concern about these terms, please let us know.

Line 224- remove “begin to”

We believe that more data are needed to fully understand the utility of time of on ART as an eligibility criterion. Our data are for a single time period and country and have the limitations stated. We are therefore more comfortable keeping “begin to” and stating our hope this begins to answer the question and that others will contribute more research to this area.

Line 2343– indicate what this high rates are.

The reference is provided if the reader would like more data on the rate.

Limitations: The failure to compare urban areas and rural areas is another limitation which the authors should acknowledge. Are the models effective the same way in both areas?

Location (urban and rural) is included in Table 1 and as noted in the text, the adjusted risk ratios included adjustment for location.

Title: The authors should modify the title to align with the results. As is, the manuscript is not showing any evidence of when patients. I suggest the title to be modified to: How soon should patients be eligible for differentiated service delivery models for antiretroviral treatment? Analysis of program data from Zambia

*Thank you – we have revised the title to include the study design
“How soon should patients be eligible for differentiated service delivery models for antiretroviral treatment? Evidence from a retrospective cohort study in Zambia”*

VERSION 2 – REVIEW

REVIEWER	Shenoi, Sheela Yale University School of Medicine, Section of Infectious Diseases
REVIEW RETURNED	30-Nov-2022
GENERAL COMMENTS	Thank you to the authors for comprehensive response and revision. All queries have been answered.